# RIPK1 in Diffuse Glioma Pathology: From Prognosis Marker to Potential Therapeutic Target

**DOI:** 10.3390/ijms26125555

**Published:** 2025-06-10

**Authors:** Leslie C. Amorós Morales, Santiago M. Gómez Bergna, Abril Marchesini, María Luján Scalise, Nazareno Gonzalez, M. Leticia Ferrelli, Marianela Candolfi, Víctor Romanowski, Matias L. Pidre

**Affiliations:** 1Instituto de Biotecnología y Biología Molecular (IBBM, UNLP-CONICET), Facultad de Ciencias Exactas, Universidad Nacional de La Plata, Consejo Nacional de Investigaciones Científicas y Técnicas, La Plata B1900, Argentina; amorosleslie@biol.unlp.edu.ar (L.C.A.M.); sgomezbergna@biol.unlp.edu.ar (S.M.G.B.); abrilmarchesini@biol.unlp.edu.ar (A.M.); lujanscalise@biol.unlp.edu.ar (M.L.S.); lferrelli@biol.unlp.edu.ar (M.L.F.); 2Instituto de Investigaciones Biomédicas (INBIOMED, UBA-CONICET), Facultad de Medicina, Universidad de Buenos Aires, Consejo Nacional de Investigaciones Científicas y Técnicas, Ciudad Autónoma de Buenos Aires C1121ABG, Argentina; gonzalez.nazareno1@gmail.com (N.G.); mcandolfi@fmed.uba.ar (M.C.)

**Keywords:** RIPK1, diffuse gliomas, proinflammatory cell death, cisplatin

## Abstract

Diffuse gliomas (DGs) are malignant primary brain tumors originating from glial cells. This study aimed to investigate the role of Receptor-interacting protein kinase 1 (RIPK1) in DG pathology. The RIPK1 mRNA expression was analyzed in DG databases from The Cancer Genome Atlas (TCGA) containing clinical, genomic, and transcriptomic information from 670 patients. Transcriptomic studies were carried out using USC Xena and R, while in vitro assays were performed with the glioblastoma human cell line U251 and the commercial RIPK1 inhibitor GSK2982772. The results showed that high RIPK1 expression was linked to a lower survival probability in patients. Additionally, the RIPK1 expression was higher in the wtIDH samples compared to that in the mIDH samples. Significant differences in the expression of genes related to cellular dedifferentiation, proinflammatory cell death pathways, and tumor-infiltrating immune cells were found between high- and low-RIPK1 expression groups. To further characterize the role of RIPK1 in DG, the effects of the RIPK1 inhibitor were evaluated, alone or combined with cisplatin, on glioblastoma cell proliferation and apoptosis. The combined treatments effectively reduced cell proliferation and increased apoptosis. The overexpression of RIPK1 was associated with a poor prognosis for DG, suggesting that RIPK1 plays a critical role in glioma pathogenesis and should be considered in therapeutic decision-making.

## 1. Introduction

Diffuse gliomas (DGs) are the most frequent malignant primary brain tumors in adults [1,2]. Despite advances in conventional therapies, most patients with DGs eventually die due to its highly invasive nature and intrinsic resistance to therapies. For this reason, it is imperative to identify new molecular targets and develop novel treatment strategies [3].

Historically, diffuse gliomas (DGs) have been classified into four grades (I to IV) based on their histological characteristics (e.g., astrocytoma, oligodendroglioma, and glioblastoma), with grade IV being the most aggressive. Among the gliomas, glioblastoma multiforme (GBM) is the most common malignant primary brain tumor in adults and is classified as grade IV. Despite current treatments, which typically consist of surgical removal of the tumor followed by chemoradiotherapy, the average life expectancy of patients is usually around 12 months [4].

Due to the histologic classification of diffuse gliomas depending on the pathology criteria, driving discrepancies in diagnosis, the World Health Organization (WHO) included distinctive genetic and epigenetic alterations to define various groups of gliomas in 2016 [5,6,7,8], and then actualized this classification in the fifth edition of the Classification of Tumors of the Central Nervous System, published in 2021 [9]. At present, the mutational status of isocitrate dehydrogenase (IDH) 1 and 2 represents the main marker for the molecular classification and prognosis of DG in adults. A mutation in the R132 residue of the active site modifies the catalytic activity of the enzyme, which converts αKG to 2-hydroxyglutarate (2-HG) [10,11,12]. The accumulation of 2-HG leads to hypermethylation-associated epigenetic reprogramming of tumor cells [11,12]. Although the IDH mutation (mIDH) is associated with a poor prognosis in some cancers [13], in patients with glioma, it is associated with a higher survival rate when compared with wild type IDH (wtIDH) patients [3,14,15].

Programmed cell death (PCD) is not only crucial in the maintenance of organism homeostasis, but also plays an important role in the progression of cancers [16]. While apoptosis has been extensively studied as an important anticancer defense mechanism, the roles of necroptosis and pyroptosis in cancer are not yet fully understood. On one hand, pyroptosis is closely related to diseases of the nervous system, infectious diseases, autoimmune diseases, cardiovascular diseases, and tumors [17]. On the other hand, given its ambiguous role in cancer biology, necroptosis has emerged as a novel target for cancer therapy, and an increasing number of compounds inducing or manipulating necroptosis are in the pipeline of development [18,19].

Receptor-interacting protein kinase 1 (RIPK1) is a master regulator of the cellular decision between pro-survival NF-κB signaling and death in response to a broad set of inflammatory stimuli [20]. Furthermore, it is involved in the formation of the PANoptosome, a multimeric protein complex that can induce pyroptosis, apoptosis, and necroptosis and activate proinflammatory cell death [21].

Various diseases, such as autoimmune disorders, inflammatory diseases, neurodegenerative conditions, and certain cancers, have been linked to RIPK1-mediated pathways, drawing the attention of various researchers. Related to gliomas, the upregulation of necroptosis-associated genes was reported in low-grade glioma and GBM when compared with normal tissue, and this was associated with a worse prognosis [22,23]. In this sense, the possibility of inhibiting RIPK1 activity has gained particular relevance, with Necrostatin-1 (Nec-1)—a natural inhibitor of necroptosis—emerging as a potential therapeutic agent for this purpose [24]. Likewise, other RIPK1 inhibitors based on small molecules have emerged, demonstrating high precision. Among them, the inhibitor GSK2982772, developed by GlaxoSmithKline (GSK), became the first RIPK1 inhibitor approved for clinical research worldwide in 2014 [25]. Although it has shown good cell membrane permeability, it demonstrated low penetration into rat brain [26]. In this context, Zhou J et al. used Nec-1 and GSK872 to inhibit RIPK1 and RIPK3, respectively, and demonstrated that necroptosis is the critical death mechanism induced by emodin, an anthraquinone compound with antitumor activities, in the U251 glioma cell line [27]. Meanwhile, in the U118 GBM cell line, researchers observed that the inhibition of RIPK1 and RIPK3 expression affected necroptosis but increased apoptosis after treatment with edelfosine [28]. Similarly, Zhang and collaborators used Nec-1 to inhibit RIPK1 expression and observed a decrease in cisplatin-induced apoptosis in the human esophageal squamous cell carcinoma cell line KYSE510, and a significant increase in cell proliferation when combining Nec-1 with cisplatin, with respect to monotherapy with the chemotherapy [29]. Moreover, in ovarian cancer, it was reported that the knockout (KO) of RIPK1 reduced cell proliferation and tumor growth in a mouse xenograft tumor model. However, RIPK1 KO attenuated the cytotoxic effect of cisplatin in vitro and in vivo, demonstrating the dual role of RIPK1 in ovarian cancer, acting as a tumor-promoting factor to maintain cancer cell proliferation or as a tumor-suppressing factor to facilitate the anticancer activity of cisplatin by the regulation of both apoptosis and necroptosis [30]. Building on these findings, in 2024, researchers developed a small-molecule proteolysis-targeting chimera (PROTAC) to inhibit RIPK1 expression and further study the role of immunogenic cell death and RIPK1 in the anticancer response. It was observed that the degradation of RIPK1 enhanced the immunostimulatory effects of radio- and immunotherapy by sensitizing cancer cells to treatment-induced TNF and interferons in several malignant cell lines [31,32].

Given all of the aforementioned information, RIPK1 appears to be an interesting gene to be evaluated as a molecular marker and a potential therapeutic target in DGs.

Therefore, in this work, we aimed to characterize the role of RIPK1 in tumor progression and the tumor immune microenvironment (TIME) of DGs through comprehensive in silico analyses of patient databases, as well as pre-clinical in vitro assays evaluating the effect of combining a commercial RIPK1 inhibitor with a chemotherapeutic agent on cell proliferation and apoptosis.

## 2. Results

### 2.1. RIPK1 Expression and Survival

In order to study the relationship between RIPK1 expression levels and the aggressivity of diffuse gliomas, the TCGA-LGGGBM (The Cancer Genome Atlas-Low-Grade Glioma and Glioblastoma Multiforme) database was employed to evaluate 670 samples from patients.

First, the samples were separated into two groups, with high and low RIPK1 expression; then, the survival probability was plotted for both groups. A significantly lower survival probability was observed in the patients with higher RIPK1 expression (Figure 1A).

For a more exhaustive analysis, the samples were filtered by the IDH status, mutated or wild type, before being divided based on RIPK1 expression. Hereby, no significant differences in survival probability were observed under conditions of either low or high expression of RIPK1 for patients with mutated IDH1 (Figure 1A). However, in those samples with wild type IDH1, the survival probability was significantly decreased in the high-RIPK1-expression group (Figure 1A).

Regarding the expression of RIPK1 in the samples derived from patients with different subtypes of glioma, our results showed higher levels of RIPK1 in the samples derived from patients with GBM and astrocytoma CNS (central nervous system) grade 2–3 (A2-3) compared with the samples of low-grade gliomas, such as oligodendrogliomas (ODs) and wtIDH gliomas grade 2–3 (OA) (Figure 1B).

To evaluate the RIPK1 expression and IDH status, the samples were divided according to them being wild type or having mutated IDH. Thus, a significant increase in the expression of RIPK1 in wtIDH in comparison with mIDH was observed (Figure 1C).

On the other hand, the samples were divided according to the patients who had received radiotherapy and those who had not, and the RIPK1 expression was studied in both cases. A higher expression of RIPK1 was observed in the samples derived from the patients treated with radiotherapy compared with the samples from the untreated patients (Figure 1D).

### 2.2. RIPK1 Differential Expression Analysis

A differential expression analysis comparing samples with high and low RIPK1 expression was performed in the Xena platform. Volcano plots of each analysis and a heatmap of differentially expressed genes were created for the wtIDH and mIDH samples (Figure 2A). Then, a gene set enrichment analysis (GSEA) was performed and an over-representation of genes involved in inflammatory pathways, TNF-ɑ signaling, and the EMT (epithelial–mesenchymal transition) pathway in both the mIDH and wtIDH samples was observed (Figure 2B).

Then, the differentially expressed genes were used to perform a Gene Ontology (GO) enrichment analysis (Figure 3A) and a transcription factor (TF) enrichment analysis (Figure 3B). This first analysis showed that in those samples with high expression of RIPK1, the pathways linked to cell dedifferentiation, inflammation, and cell death were upregulated. Moreover, we observed an over-representation of transcription factors like SPI1, STAT3, IRF, and RELA involved in the regulation of the expression of different sets of genes related to proliferation, cell dedifferentiation, inflammation, and cell cycle deregulation.

### 2.3. Apoptosis

The differential expression of apoptosis-related genes was analyzed in both the high and low RIPK1 expression groups. Additionally, the samples were divided by the IDH1 status and the correlation between the RIPK1 expression and apoptosis-related genes was studied. It was observed that BAX, CASP3, CASP8, TP53, CYCS, and NFκB1 had significantly higher levels in the group with high RIPK1 expression compared with those with lower levels of RIPK1 (Figure 4A).

### 2.4. Necroptosis and Pyroptosis

The genes related to necroptosis were analyzed under conditions of either low or high RIPK1 expression. The results showed higher levels of RIPK3, MLKL, CYLD, CASP8, and BIRC3 in those samples with high RIPK1 expression while no significant differences were observed for CYLD. Also, the correlation between the necroptosis-related genes and RIPK1 expression was studied in three different groups, samples with mutated IDH1, those samples with wild type IDH1, and a group that contained both statuses, showing a positive correlation between most of the studied genes and RIPK1 expression for all groups (Figure 4B).

A similar analysis was performed to study pyroptosis. In this case, BAX, CASP1, CASP3, CASP4, CASP8, GSDMD, GZMA, IL6, and IL1β were evaluated and a higher expression was observed; a strong positive correlation between all of the previously mentioned genes in the samples with high RIPK1 expression compared with those with low RIPK1 expression was also found (Figure 4C).

### 2.5. Proliferation

Cell proliferation is a vital process for tumor progression. Thus, the transcriptomic analysis was repeated, evaluating genes linked to this process under conditions of low and high RIPK1 expression. In addition, the correlation between these genes and RIPK1 expression was studied, grouping the samples by IDH1 status. The results showed a higher expression of MAPK8, MAP2K1, APC, KRAS, and HRAS and a lower expression of JUN, STAT3, NFKB1, and CTNNB1 in the group with low RIPK1 expression in both the mIDH and wtIDH samples, while higher levels of GSK3B and RAF1 were observed in the group of low RIPK1 in the mIDH samples. The same was observed for MAPK3, PTPN11, and AXIN2 for the wtIDH1 samples (Figure 5A).

### 2.6. RIPK1 and EMT

In the context of neoplasias, the epithelial–mesenchymal transition (EMT) is associated with tumor initiation and invasion [33]. The study of the correlation between RIPK1 expression and the expression of EMT-related genes was carried out in samples with mIDH or wtIDH. It was observed that the samples with high RIPK1 expression presented significantly higher levels of the majority of the evaluated genes for both the wtIDH and mIDH samples (Figure 5B).

### 2.7. RIPK1 Expression and Immune Cell Infiltration

Given the potential role of RIPK1 in glioma immunity due to its role in necroptosis and pyroptosis, we performed a meta-analysis using previously validated immune genetic signatures of tumor-infiltrating immune cells [34,35]. We found that the expression of genetic signatures of different lymphocyte populations, i.e., helper, cytotoxic, memory, and regulatory T cells, was upregulated in the mIDH gliomas with high RIPK1 expression levels (Figure 5A). Similarly, the expression of the gene signatures of antigen-presenting cells, i.e., dendritic cells (DCs) and macrophages, was upregulated. The same effect was observed regarding neutrophils and NK cells (Figure 6A).

For the wtIDH gliomas, we observed that the expression of the genetic signatures of the macrophages and neutrophils was upregulated in the samples with high RIPK1 expression levels (Figure 6B). Regarding the expression of the genetic signatures of different lymphocyte populations, we only observed an increase in memory T cells. However, we observed a significant decrease in the expression of genetic signatures of Tregs in the wtIDH gliomas with high RIPK1 expression levels (Figure 6B).

On the other hand, we performed a spatial transcriptomic analysis based on tissues deposited on STOmicsDB (Figure 7).

First, we evaluated the RIPK1 and TNFα receptor expression in both normal brain tissue and GBM biopsies and we found higher expression levels of both transcripts and a greater distribution in the GBM tissues, with detectable expression in 25% and 81% of the clusters, respectively (Figure 7A).

Then, we compared the transcript levels of RIPK1 and relevant genes involved in proinflammatory cell death pathways and immune infiltrating cell signatures in each cluster of the GBM tissue. In Figure 7B, it can be observed that clusters 1, 2, 3, 5, 7, 10, 11, 12, and 14 present higher levels of RIPK1 expression. However, not all of these clusters present greater immune infiltration based on the expression of characteristic gene signatures. Interestingly, clusters 10–14 presented a higher expression of immune cell gene signatures such as CD4, CD8, CD33, CD28, CD27, CD14, and CD68. Three of these four clusters exhibit high expression of RIPK1 too. Finally, we found a greater expression of macrophage gene signatures (CD68) compared to other immune cell signatures (Figure 7C).

### 2.8. RIPK1 Inhibition Combined with Chemotherapy Reduces Cell Growth and Induces Apoptosis In Vitro

Due to the critical role of RIPK1 in the programmed necrosis pathway, several selective small-molecule inhibitors of its kinase activity have been developed for potential clinical use. In particular, GSK2982772 is an effective ATP competitive RIPK1 inhibitor, currently being evaluated through different clinical trials for use in other CNS diseases [36,37].

With the aim of further studying the role of RIPK1 in the pathogenesis of DG, pre-clinical in vitro assays were performed using the human glioblastoma cell line U251 and the highly selective commercial RIPK1 inhibitor, GSK2982772.

First, we treated U251 cells with the RIPK1 inhibitor and no significant differences were observed in cell proliferation compared to the negative control (Figure 8A). Subsequently, we studied the potential effect on cell proliferation of a combined therapy using GSK2982772 and cisplatin (cis-diamminedichloroplatinum(II) or CDDP), one of the most widely prescribed anticancer drugs [38].

We treated U251 cells with different concentrations of the RIPK1 inhibitor and 5 μM of CDDP. After 72 h, we observed a significant decrease in cell proliferation in cells treated with both drugs compared to those treated with CDDP alone (Figure 8B).

Moreover, we decided to study the effect of RIPK1 inhibition on sensitivity to CDDP. For this, we added a step of 24 h sensitization with GSK2982772 prior to treatment with CDDP, using a reduced drug concentration. We observed that for the 1 μM concentration of CDDP, prior sensitization with the RIPK1 inhibitor significantly impacted the cell proliferative capacity compared to the control with CDDP monotherapy (Figure 8C).

Additionally, we evaluated the apoptosis levels using TUNEL. The results revealed higher levels of apoptosis induction when cells were sensitized with GSK2982772 and subsequently treated with CDDP (Figure 8D).

## 3. Discussion

Diffuse gliomas are among the most common and deadly types of primary brain tumors [1,2]. Patients with mIDH gliomas typically present a better prognosis [39], but they often evolve to higher grades. In contrast, wtIDH gliomas are usually present as glioblastomas (GBMs), the most common and clinically aggressive type, with a median survival of only 12 to 15 months [40].

Although gliomas are among the most deeply genetically characterized of all tumor types thanks to the efforts of multicenter research consortia such as The Cancer Genome Atlas (TCGA) [41], the last addition to the treatment repertoire was temozolomide (TMZ), 17 years ago [40]. Thus, transcriptomic studies focused on understanding the pathogenesis of gliomas have become particularly relevant.

Here, we studied the role of RIPK1 on DG pathology. In contrast to a previous analysis, where RIPK1 expression did not correlate significantly with a poorer overall survival and disease-free survival [22], we observed a higher probability of survival in the patients with a lower RIPK1 expression. More specifically, only the wtIDH patients showed significant differences when the samples were stratified by IDH1 mutational status. Furthermore, we observed significant differences in the RIPK1 expression levels in relation to the histological classification of DGs. The same occurred when we analyzed the expression of RIPK1, classifying the patients according to their IDH1 mutational status. Together, this information suggests that more aggressive tumors exhibit higher RIPK1 expression. These results are consistent with previously reported data, since RIPK1 is a master regulator of necroptosis, and necroptosis-pathway-associated genes are unfavorable prognostic markers in GBM [22,23]. Additionally, we observed higher levels of RIPK1 expression in those patients previously treated with radiotherapy, suggesting that RIPK1 could be involved in some mechanism triggered in response to treatment, such as inflammation induced by radiotherapy [42].

From our differential expression and GO analyses, we observed a strong positive correlation between RIPK1 and the genes involved in cell death pathways, especially pyroptosis and necroptosis.

One of the most important characteristics of GBM is hyper-vascularization, associated with resistance to anti-angiogenic therapy (AATx). Wei and collaborators reported that the inhibition of TNFα secreted by glioma-associated macrophages (GAMs) inhibited the activation of endothelial cells, improving survival in a mouse glioma model and prolonging the durability of the response to AATx [43]. In this work, we observed an over-representation of certain transcription factors involved in regulating the expression of genes related to TNFα in samples with high RIPK1 expression.

These results, together with those related to prognosis, suggest that the deregulation of some of these pathways could contribute to the lower survival probability observed in patients with high RIPK1 expression. This is consistent with the results previously reported by Lin et al. [44] and Zhou et al. [23], in which they demonstrate that greater activation of inflammatory cell death pathways correlates with a worse clinical prognosis. At the same time, Cao et al. [45] reported that radiotherapy treatment induces greater activation of pyroptosis and consequently more inflammation, consistent with our findings.

Although temozolomide is currently the most used therapeutic agent for GBMs, the effectiveness of this treatment remains low [46]. In this context, some studies have reported combining temozolomide and CDDP in clinical trials, showing better results with respect to TMZ monotherapy [47]. Recently, Zou and collaborators developed nanoparticles carrying both TMZ and CDDP to avoid the blood–brain barrier, with promising therapeutic potential [48]. Concerning our pre-clinical results, we demonstrated that the combination of the RIPK1 kinase inhibitor and CDDP significantly decreased the proliferation of human GBM cells. Interestingly, the effect on cell proliferation was even greater when cells were sensitized for 24 h with the inhibitor before treatment with CDDP. Moreover, sensitization with GSK2982772 prior to the addition of 1 μM of CDDP induced higher levels of apoptosis in U251 cells. The rationale of using CDDP relies on the fact that cisplatin induces both DNA damage and ROS and this is associated with more inflammatory cell death. These results are consistent with the work of Melo-Lima and collaborators [28], who observed that the combination of a RIPK1 inhibitor with edelfosine (which induces inflammatory cell death when supplied alone), could redirect death pathways, resulting in higher levels of apoptosis and lower inflammation. However, more in vitro and in vivo assays using mIDH models are necessary for a deeper characterization of RIPK1 as a molecular target.

In relation to the immune response, different authors point out that the IDH mutation in glioma is associated with a reduction in immunological infiltrates, leading to a better prognosis in patients [49,50]. In fact, Gonzalez et al. [3] demonstrated that mIDH gliomas have less immunological infiltration than wtIDH tumors and a greater relative abundance of NK cells [51,52].

Regarding our findings, we observed that higher RIPK1 expression correlates with higher immune infiltration in both mIDH and wtIDH tumors.

In particular, the high-RIPK1 wtIDH gliomas showed an upregulation of macrophages, neutrophils, and memory T cells, while the Treg and NK cell gene signatures were significantly lower than those of the low-RIPK1 tumors. The accumulation of macrophages and neutrophils characterizes the unsolved chronic inflammation involved in the pro-tumorigenic effect of the TIME in gliomas [53], and thus, it is coherent to find that these gene signatures are upregulated in tumors with such poor prognosis as that of wtIDH gliomas [54,55].

However, in the spatial transcriptomic analysis, we found that, despite RIPK1 and TNFα receptor expression being higher in GBMs compared to normal tissue, their spatial distribution seemed to be uniform through the sample. In this way, we could not identify relevant regions of correlation between the RIPK1 and immune cell gene signatures.

Apart from immunological cells, some studies have highlighted the relevance of astrocytes in tumor progression. Increased astrocyte reactivity, triggered by direct interactions with glioma cells, can alter the tumor microenvironment by producing inflammatory factors such as IFNα and TNFα, among others [56,57].

It is well known that RIPK1 is involved in the inflammatory response and necroptosis [20,22,58]. Furthermore, our analysis demonstrated a higher correlation with genes involved in pyroptosis. Several studies have reported that pyroptosis may facilitate a supportive tumor microenvironment [44,59]. In this direction, RIPK1 could play a role in sustaining inflammation through the regulation of proinflammatory death pathways collaborating with glioma pathogenesis.

Altogether, our results suggest that RIPK1 plays a crucial role in glioma progression and pathogenesis. They also highlight the importance of considering RIPK1 expression levels in treatment decision-making and the development of novel therapeutic strategies.

## 4. Materials and Methods

### 4.1. Patient Cohort

For clinical and transcriptomic analysis, the TCGA-LGGGBM (Low-Grade Glioma and Glioblastoma) database was consulted and analyzed on different platforms. Patient attributes are summarized in Table 1.

### 4.2. Glioma Patient Samples

Public datasets containing clinical, genomic, and transcriptomic information from patient samples were used. In all cases, the TCGA-LGGGBM database (N = 670) was studied.

The UCSCXena platform and R were employed to analyze the data. Samples corresponding to DG were filtered, classified according to IDH status and the expression of RIPK1 and other genes was evaluated.

### 4.3. RIPK1 Expression and Clinical Attributes

For these analyses, the glioma database and the UCSCXena [60] platform were employed. RIPK1 was used as a query and the samples were divided into two groups according to the median value of expression: high expression of RIPK1 (>9.094) and low expression of RIPK1 (≤9.094). The correlation between RIPK1 expression levels and several clinical attributes of interest was assessed.

### 4.4. Survival Plots

Samples corresponding to different types of glioma were filtered and the expression of RIPK1 was evaluated. For the analysis, samples were divided into two groups according to the median value, and survival was plotted for each group.

### 4.5. Transcriptomic Analysis

The mRNA expression data in log_2_(x+1)-transformed RSEM normalized counts obtained with Illumina HiSeq2000 RNA (University of North Carolina, Chapel Hill, NC, USA) sequencing platform were used.

Differential expression analysis was performed in the USCSXena platform, using the Limma-Voom method, taking as differentially expressed genes those with adjusted *p*-values < 0.05 and |log2FC| > 1.5.

For different pathway correlation analyses, the UCSCXena platform was used, normalizing counts as z-scores relative to all samples. Differentially expressed genes were visualized in volcano plots and heat maps generated using R.

For spatial transcriptomic analysis, the dataset STDS0000040, corresponding to a glioblastoma multiforme (GBM) tissue sample, was obtained from StomicsDB [61]. Fresh frozen human GBM tissue was acquired by 10× Genomics from BioIVT Asterand (https://support.10xgenomics.com/spatial-gene-expression/datasets/1.2.0/Parent_Visium_Human_Glioblastoma, accessed on 14 May 2025). The tissue was embedded and cryosectioned according to the Visium Spatial Protocols—Tissue Preparation Guide (Demonstrated Protocol CG000240). Tissue sections of 10 µm thickness were placed on Visium Gene Expression slides, then fixed and stained following the Methanol Fixation, H&E Staining & Imaging for Visium Spatial Protocols (CG000160). Clusters and tissue gene expression were visualized using the StomicsDB web server (https://db.cngb.org/stomics/datasets/STDS0000040/summary, accessed on 14 May 2025). For cluster expression analysis, processed data were downloaded from StomicsDB. Z-score normalized expression values were calculated for each gene across clusters

### 4.6. Enrichment and Functional Analysis

Differentially expressed genes were used to perform a GO, pathway, and TF enrichment analysis with Enrichr extension in the USCSXena platform. For GO enrichment, GO terms 2023 were used, while ENCODE and ChEA Consensus TFs from ChIP-X were used for TF enrichment analysis.

In parallel, differentially expressed genes were used to perform a GSEA with blitzGSEA extension available in the USCSXena platform. The MSig_DB Hallmark gene sets (H) from the human collection were used for the analysis.

Statistical significance was considered as an adjusted *p*-value < 0.05.

### 4.7. Meta-Analysis of Immune Gene Signatures in mIDH and wtIDH Glioma Biopsies

A total of 521 patients with gliomas from the TCGA-LGGGBM cohort were analyzed in this study. We stratified patients according to the mutational status of IDH into two groups: mIDH and wtIDH patients. Samples were classified into two groups according to RIPK1 mRNA expression levels, RIPK1low and RIPK1high, using median expression values as cut-offs. Normalized expression values were log2-transformed. Genes were grouped into 8 immune cell type signatures (Cytotoxic CD8, Helper T cells, Dendritic cells, NK, Macrophages, Neutrophils, Memory T cells, and Treg), derived from Gonzalez et al. [3]. ESTIMATE scores were obtained from the cBio Cancer Genomics Portal [62,63]. Gene signatures for each immune cell type were obtained from previously published data [34,35].

### 4.8. Glioblastoma Multiforme (GBM) Cells

Human GBM U251 cells were cultured in Dulbecco’s modified Eagle’s medium (DMEM, Gibco, Rockville, MD, USA) plus 10% fetal bovine serum (FBS) in monolayer culture, at 37 °C in a humidified atmosphere of 5% CO_2_.

### 4.9. Tumor Cell Growth

Three different experiments were performed using the inhibitor of RIPK1 (GSK2982772, MedChemExpress, Monmouth Junction, NJ, USA). First, U251 cells were seeded into 96-well plates at a density of 2 × 10^3^ per well. After 24 h, GSK2982772 concentrations ranging from 8 to 48 nM in DMEM+ 10% FBS were added to the wells as a unique treatment. Secondly, U251 cells were incubated under the same conditions but in a combined treatment with GSK2982772 and 5 μM of cisplatin (CDDP or cis-diamminedichloroplatinum(II)). Finally, we performed a sensitization assay, incubating cells with GSK2982772 for 24 h and then with 1 μM of CDDP. Control wells were treated with the maximum volume of the vehicle. For the three experiments, medium was removed after 72 h and plates were fixed with 4% paraformaldehyde for 15 min, stained with a 0.5% crystal violet solution for another 15 min, and washed with rinse water. The dye in the stained cells was dissolved in 10% methanol–5% acetic acid solution (*v*/*v*). Absorbance at 595 nm was read in an automated plate spectrophotometer (TECAN Infinite 200 PRO, Crailsheim, Germany).

### 4.10. Apoptosis by TUNEL

Apoptotic cells were detected using terminal deoxynucleotidyl transferase dUTP nick end labeling (TUNEL) using the DeadEnd Fluorometric TUNEL System (Promega, Madison, WI, USA) and DNA was stained with DAPI. Briefly, U251 cells were cultured on glass coverslips. After 24 h, cells were incubated with 16 nM of GSK2982772 alone or in combination with 1 μM of CDDP for 48 h. Cells were visualized in a fluorescent light microscope (BioTek Cytation 5, Agilent, Santa Clara, CA, USA). The percentage of TUNEL + cells was determined by analyzing the images obtained with ImageJ Fiji software 2.0.

### 4.11. Statistical Analysis

Expression of RIPK1 in samples of TCGA-LGGGBM vs. different DG subtypes: n = 1106; Kruskal–Wallis test followed by the Dunn test with the *p*-value corrected by the Bonferroni method was used. Expression of RIPK1 in samples of TCGA-LGGGBM vs. radiotherapy treatment: n = 701; Welch’s *t*-test. Survival plots: n = 702; log-rank test. RIPK1 expression vs. proteins of different pathways: n = 702 (n = 351 high expression of RIPK1; n = 351 low expression of RIPK1); Welch’s *t*-test. Correlation analysis: Pearson’s correlation coefficient. Meta-analysis of immune gene signatures in mIDH (n = 311) and wtIDH (n = 210): Mann–Whitney test. Statistical significance for all in vitro assays was assessed using one-way ANOVA with Dunnett’s multiple comparison as a posteriori test or Student’s *t*-test. Statistical significance was set at * *p* < 0.05.

## Figures and Tables

**Figure 1 ijms-26-05555-f001:**
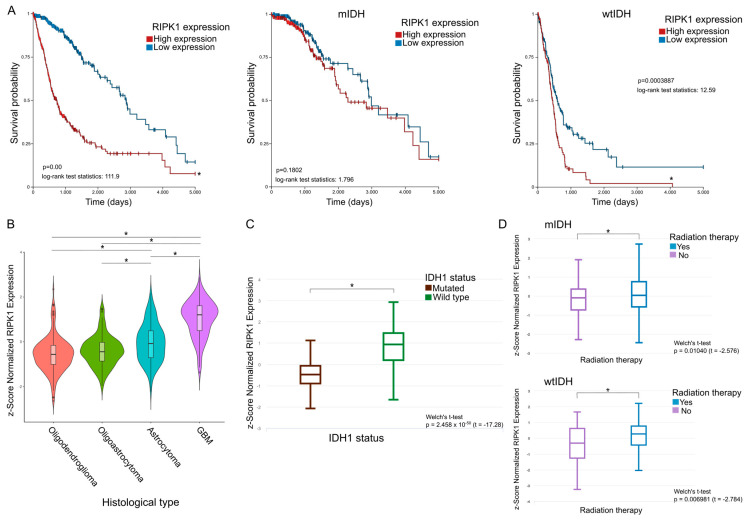
(**A**) Survival plots. DG samples were filtered from TCGA, and the expression of RIPK1 was evaluated. In the analysis, the samples were divided into two groups according to the median value of expression: high expression of RIPK1 and low expression of RIPK1. Survival was plotted for each group. Data were plotted unclassified (left) or classified into mIDH (center) or wtIDH (right). Log-rank test * *p* < 0.05. (**B**) Differential expression of RIPK1 between DG tumor subtypes. Kruskal–Wallis test followed by the Dunn test with the *p*-value corrected by the Bonferroni method was carried out. * *p* < 0.05. (**C**) Differential expression of RIPK1 in mIDH and wtIDH gliomas. Welch’s *t*-test. * *p* < 0.05. (**D**) Differential expression of RIPK1 between samples of patients who received radiotherapy and those who did not. Samples were classified into mIDH (top) and wtIDH (bottom). Welch’s *t*-test. * *p* < 0.05.

**Figure 2 ijms-26-05555-f002:**
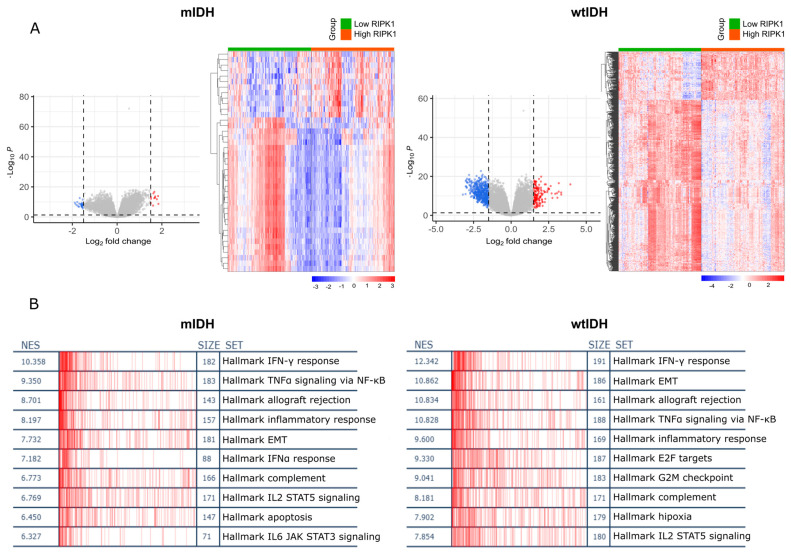
(**A**) A volcano plot and heatmap of differentially expressed genes. The volcano plot shows the distribution of genes based on log₂ FC and -log₁₀ (P-adjusted). Red dots represent significantly upregulated genes, blue dots represent significantly downregulated genes, and gray dots indicate non-significantly differentiated genes. Heatmap of differentially expressed genes across samples. Expression values are scaled by row (z-score), with red indicating higher expression and blue indicating lower expression. (**B**) Table displays the top 10 enriched gene sets with their normalized enrichment scores (NESs) and the distribution of hits relative to the gene ranking of the signature.

**Figure 3 ijms-26-05555-f003:**
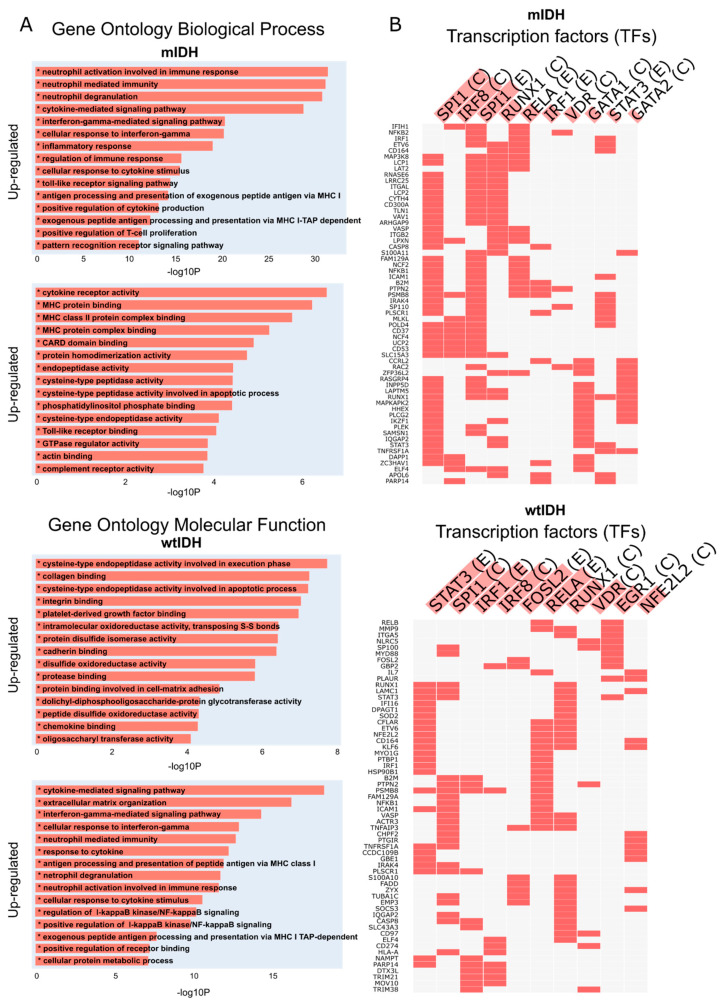
Gene ontology and transcription factor (TF) enrichment. (**A**) Plots show the canonical categories significantly over-represented (enriched) by the DEGs for the main categories of two ontologies (GO: biological process and GO: molecular function) for mIDH samples (above) and wtIDH samples (below). The information was obtained from The Cancer Genome Atlas Pan-Cancer-Xena database. DEG, differentially expressed gene. (**B**) FET (Fisher’s exact test) *p*-value of top 10 TFs. In columns are the enriched terms, while input genes are in rows. Red squares in the matrix indicate if a gene is associated with a term. (E) and (C) represent TFs that are present in ENCODE and ChEA databases, respectively.

**Figure 4 ijms-26-05555-f004:**
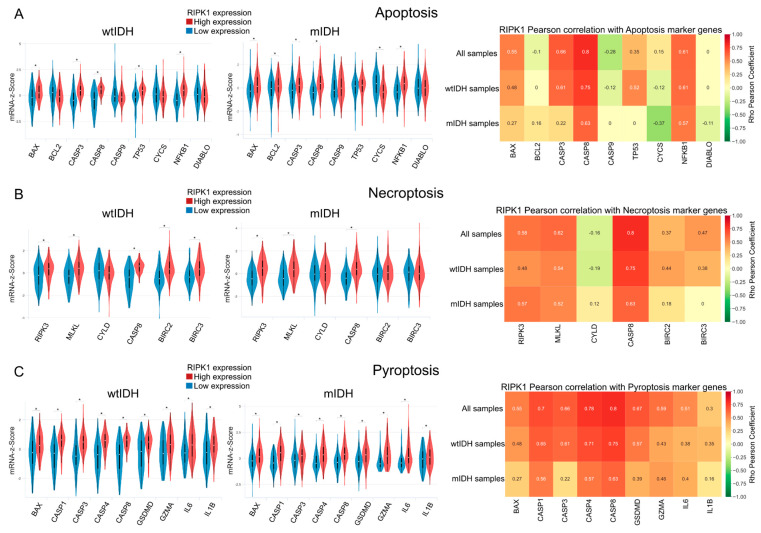
Transcriptomic analysis of PCD pathways. (**A**) Apoptosis, (**B**) necroptosis, (**C**) pyroptosis. In each panel, left: violin plots of the expression (z-score) of genes involved in the respective pathway under conditions of high and low RIPK1 expression. Samples were classified as mIDH and wtIDH and then plotted. Welch’s *t*-test * *p* < 0.01. Right: Correlation between RIPK1 expression and genes involved in the respective pathway separated in all samples of mIDH and wtIDH tumors. Numbers in cells indicate the Pearson correlation index in statistically significant comparisons.

**Figure 5 ijms-26-05555-f005:**
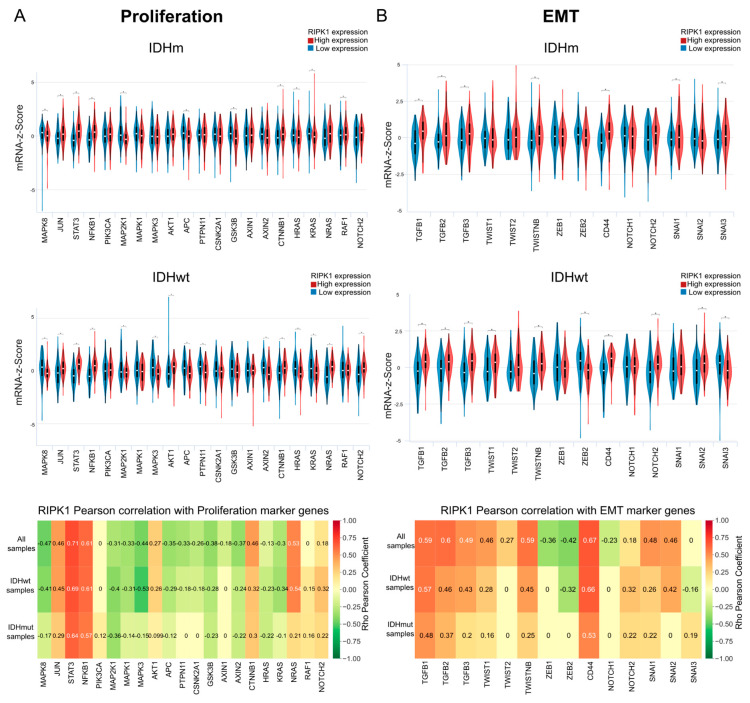
Transcriptomic analysis of (**A**) proliferation and (**B**) EMT. Samples were classified into mIDH and wtIDH and violin plots of the expression (z-score) of genes involved in the respective pathway were made under conditions of high and low RIPK1 expression (Welch’s *t*-test * *p* < 0.01). Above: Correlation between RIPK1 expression and genes involved in the respective pathway separated in all samples, mIDH and wtIDH tumors. Numbers in cells indicate the Pearson correlation index in statistically significant comparisons.

**Figure 6 ijms-26-05555-f006:**
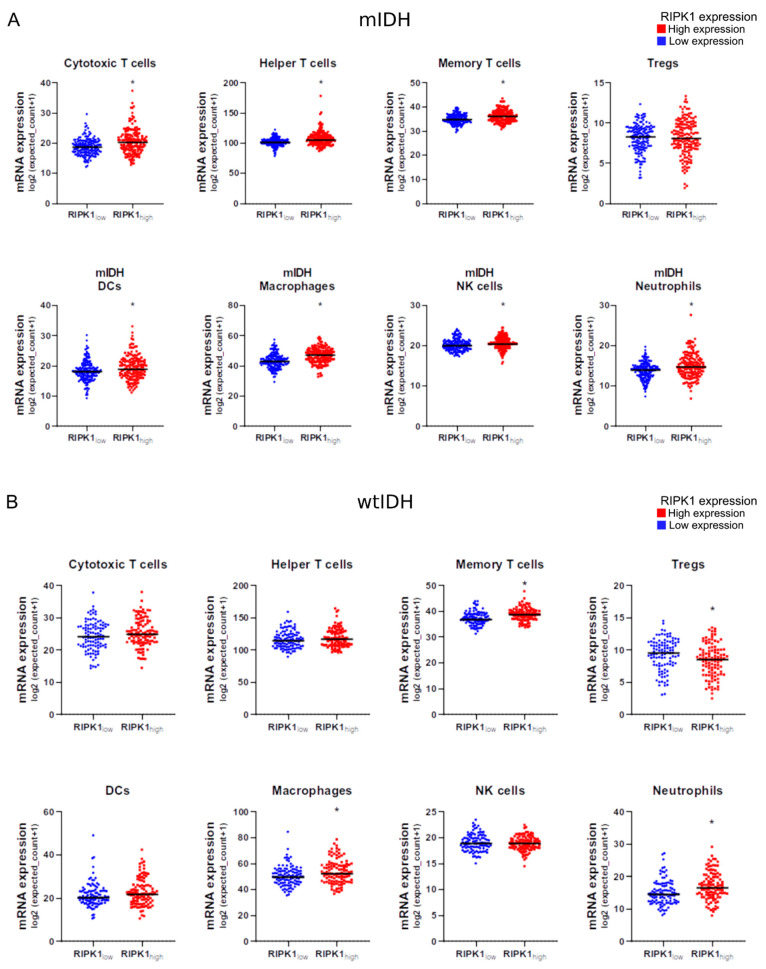
Meta-analysis of immune gene signatures in mIDH and wtIDH glioma biopsies. We evaluated the gene signatures that characterize immune cell populations in transcriptomic data of glioma biopsies deposited in The Cancer Genome Atlas. Data were stratified according to the 2021 WHO classification in (**A**) mIDH (*n* = 311) and (**B**) wtIDH gliomas (*n* = 210). Patients in each group were classified according to RIPK1 mRNA expression levels into two groups (RIPK1low and RIPK1high). *, *p* < 0.05; Mann–Whitney test.

**Figure 7 ijms-26-05555-f007:**
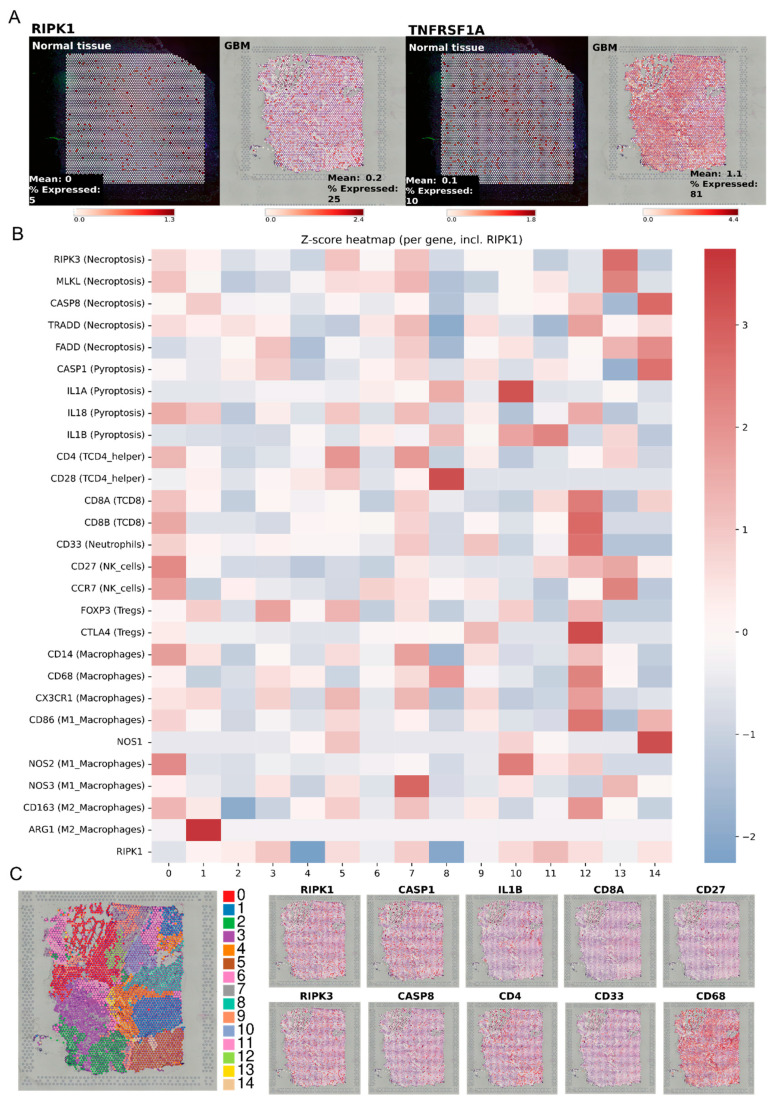
Gene expression analysis in spatial transcriptomic sample. We evaluated gene expression of death-related markers and immune cell markers in GBM tissue. (**A**) Expression of RIPK1 and its receptor TNFRSF1A in GBM samples and normal tissue. (**B**) Normalized mean expression by z-score of selected genes and tissue clusters. Z-score was calculated among all clusters per gene. (**C**) Tissue cluster localization and selected gene expression in the tissue.

**Figure 8 ijms-26-05555-f008:**
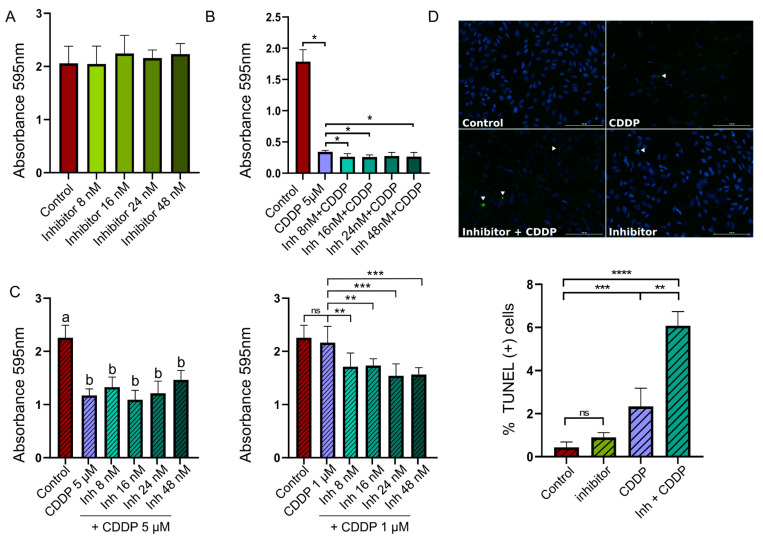
Pre-clinical validation of RIPK1 inhibition in combination with chemotherapy on an in vitro GBM experimental model. (**A**) GSK2982772 did not affect GBM cell growth. U251 cells were treated with four different concentrations of the inhibitor of RIPK1. (**B**) Combined treatment of GSK2982772 + cisplatin (CDDP) (5 μM) inhibits GBM cell growth. U251 cells were treated with different concentrations of the inhibitor of RIPK1 and 5 μM of CDDP. 72 h after treatment, cells were fixed, and cell growth was measured by absorbance at 595 nm. (**C**) Sequential therapeutic scheme with GSK2982772 sensibilization followed by CDDP inhibits GBM cell growth. U251 cells were incubated in presence of different concentrations of GSK2982772 for 24 h before being treated with 5 μM (left panel) or 1 μM of CDDP (right panel). A total of 48 h after the addition of CDDP, cells were fixed, and cell growth was measured by absorbance at 595 nm. Different letters (a or b) above the columns indicate significant difference between groups (*p* < 0.001). (**D**) Combined treatment of inhibitor + CDDP (1 μM) induces apoptosis in GBM cells. U251 cells were incubated with the inhibitor or RIPK1 (16 nM) for 24 h. Then, 1 μM of CDDP was added to the respective wells and the quantitative analysis of apoptotic cell death was performed using the TUNEL assay. In all cases, “control” refers to cells that were not treated with GSK2982772 or CDDP. ANOVA followed by Dunnett’s test, * *p* < 0.05, ** *p* < 0.01, *** *p* < 0.005, **** *p* < 0.001, “ns” means no significant differences.

**Table 1 ijms-26-05555-t001:** Patient cohort. Table 1 shows principal clinical attributes corresponding to the TCGA-LGGGBM database used in this work. The number in each cell indicates the quantity of patients with clinical attributes specified in column one.

Clinical Attribute		Database: TCGA—LGGGBM
Total of Patients		670
RIPK1	High expression	335
	Low expression	335
IDH status	wtIDH	257
	mIDH	404
	Undefined	9
Age at initial pathologic diagnosis	≤50	393
	>50	275
	Undefined	2
Histological classification	Oligoastrocytoma	130
	Oligodendroglioma	191
	Astrocytoma	324
	GBM	153
Radiotherapy	Yes	408
	No	200
	Discrepancy	1
	Undefined	61
Additional pharmaceutical therapy	Yes	134
	No	102
	Undefined	434
Additional surgery	Yes	56
	No	62
	Undefined	552

## Data Availability

The data are available upon request (mlpidre@biol.unlp.edu.ar).

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
