# Peer review of "RIPK1 in Diffuse Glioma Pathology: From Prognosis Marker to Potential Therapeutic Target"

_ijms, 2025, doi:10.3390/ijms26125555_

Round 1
Reviewer 1 Report
Comments and Suggestions for Authors
Comments to the Authors
Summary
This manuscript presents an integrative bioinformatic and limited experimental analysis of RIPK1 expression in diffuse gliomas, highlighting its association with poor prognosis—particularly in IDH-wildtype glioblastomas—and proposing it as a potential therapeutic target. The authors leverage TCGA datasets and perform basic validation using a RIPK1 inhibitor in glioma cells. The study is of interest due to the proposed stratification by IDH mutation status, which adds clinical relevance. However, the work lacks novelty, due to the substantial body of existing literature already exploring RIPK1-related pathways in gliomas (particularly glioblastoma), and requires substantial experimental support to validate the bioinformatic predictions.
General Comments
The manuscript addresses an important topic—the role of RIPK1 in glioma biology—but falls short in terms of scientific novelty and experimental depth. While the stratified analysis based on IDH status adds a relevant clinical dimension, many of the associations described (e.g., RIPK1 with immune infiltration, cell death pathways) have already been reported. The study's reliance on in silico analyses, with only minimal in vitro data, limits its translational impact.
Suggestion: To strengthen the manuscript, the authors should explicitly clarify how their results offers new insights or improves upon previous studies. I recommend the inclusion of experimental validation to corroborate the computational predictions.
Introduction comments (line 33-77)
The introduction is generally well-written and clearly outlines the study's objectives. However, it lacks a comprehensive summary of prior research investigating the association between RIPK1 expression, glioblastoma progression, and patient survival. Key studies—such as those by Dong et al. [30], Zhou et al. [31], and Melo-Lima et al. [36]—which discuss necroptosis-related gene signatures and RIPK1 activation in gliomas, are referenced only later in the Discussion and should be integrated into the Introduction to better describe the rationale for the study.
Additionally, important literature on the therapeutic potential of RIPK1 inhibition is absent. Several studies have evaluated RIPK1 inhibitors, alone or in combination with agents like cisplatin, and demonstrated their efficacy in promoting apoptosis and suppressing tumor proliferation in glioblastoma and other cancers (Pol et al., 2024; Zhou et al., 2020; Zheng et al., 2020; Zhang et al., 2021).
Suggestion: Incorporating these references would strengthen the scientific foundation of the manuscript and better support the proposed therapeutic relevance of RIPK1.
- Pol JG et al., RIPK1 inhibition in malignant cells potentiates immunotherapy and radiotherapy outcome. Oncoimmunology. 2024;13(1):2425465.
- Zhou J et al., Emodin induced necroptosis in the glioma cell line U251 via the TNF-α/RIP1/RIP3 pathway. Invest New Drugs. 2020;38(1):50–59.
- Zheng XL et al., RIP1 promotes proliferation through G2/M checkpoint progression and mediates cisplatin-induced apoptosis and necroptosis in human ovarian cancer cells. Acta Pharmacol Sin. 2020;41(9):1223–1233.
- Zhang Y et al., RIPK1 contributes to cisplatin-induced apoptosis of esophageal squamous cell carcinoma cells via activation of JNK pathway. Life Sci. 2021;269:119064.
Methods comments (line 289-385)
The cohort size (>600 patients) is appropriate for database-driven analyses and provides sufficient statistical power to support the bioinformatic conclusions, including subgroup identification, biomarker discovery, and predictive modelling.
The only experimental validation presented involves the use of the RIPK1 inhibitor GSK2982772 to assess tumor cell growth and apoptosis under three conditions (no treatment, GSK2982772 alone, and GSK2982772 combined with 5 μM cisplatin). These assays are well executed and clearly described. However, the use of a single cell line, the absence of in vivo validation, and the lack of additional functional assays significantly limit the experimental robustness and translational relevance of the findings.
Suggestion: The authors should include one IDH-mutant and one IDH-wildtype glioma line, improving the relevance of the experimental work.
Results and Discussion comments
- Association of RIPK1 expression with de-differentiation, cell death pathways, and tumor-infiltrating immune cells (line 106-175)
The transcriptomic bioinformatic analysis confirms the association between high RIPK1 expression and pro-inflammatory and de-differentiation programs. However, these associations have already been reported in previous studies, and the current analysis—based on publicly available datasets—does not offer significant novel findings. Similarly, the correlations between RIPK1 and cell death pathways are not novel, having been described in previous studies, and are based on transcriptomic data without experimental validation. The integrated analysis of immune-related pathways is well conducted, but again, not supported by experimental evidence. Given that RIPK1’s role in immunity is already described, the novelty of all these findings is limited.
Suggestion: The inclusion of spatial transcriptomic analysis or validation in patient-derived tissues of glioblastoma would improve the association of RIPK1 expression with factors linked to activation of these pathways, providing crucial experimental validation in the tumor context.
- Association of RIPK1 expression with patients survival (line 79-105, and line 183-210)
The in vitro data show that RIPK1 inhibition—particularly in combination with cisplatin—reduces cell proliferation and enhances apoptosis in glioblastoma cells. However, as previously reported, the use of a single cell line and the absence of in vivo validation significantly reduce the translational impact of these findings.
Suggestion: These limitations should be clearly acknowledged and critically discussed.
Novelty
However, the study presents two original contributions that should be emphasized more by the authors:
- Stratification of RIPK1 expression by IDHwt versus IDHmut GBM
The finding that RIPK1 expression is higher in IDHwt gliomas compared to IDHmut tumors is important. This stratified analysis adds clinical relevance because IDHwt gliomas are biologically more aggressive.
- Prognostic relevance of RIPK1 expression in IDHwt versus IDHmut GBM
The observed association between high RIPK1 expression and poorer overall survival is notable in IDHwt gliomas, aligning with their more aggressive nature. This distinction between IDHwt and IDHmut tumors is the main novel aspect of the study.
Suggestion: These results should be discussed in the context of prior literature, such as Dong et al., which reported that in glioblastoma (GBM), RIPK1 expression did not correlate significantly with poorer overall and disease-free survival. The differences in RIPK1’s prognostic significance may be explained by the lack of IDH wild-type vs. mutation stratification in previous studies. Clarifying why RIPK1 prognostic impact is more pronounced in IDHwt gliomas would enhance the scientific impact of the manuscript.
Figures and Tables
The figures and tables are well executed and clearly present the results of the proposed analyses.
Suggestion: in Supplementary Table 2: Add information on patient cohorts used in the survival analysis, including age, sex, IDH status, and treatment regimen.

Author Response
Comments to the Authors
Summary
This manuscript presents an integrative bioinformatic and limited experimental analysis of RIPK1 expression in diffuse gliomas, highlighting its association with poor prognosis—particularly in IDH-wildtype glioblastomas—and proposing it as a potential therapeutic target. The authors leverage TCGA datasets and perform basic validation using a RIPK1 inhibitor in glioma cells. The study is of interest due to the proposed stratification by IDH mutation status, which adds clinical relevance. However, the work lacks novelty, due to the substantial body of existing literature already exploring RIPK1-related pathways in gliomas (particularly glioblastoma), and requires substantial experimental support to validate the bioinformatic predictions.
General Comments
The manuscript addresses an important topic—the role of RIPK1 in glioma biology—but falls short in terms of scientific novelty and experimental depth. While the stratified analysis based on IDH status adds a relevant clinical dimension, many of the associations described (e.g., RIPK1 with immune infiltration, cell death pathways) have already been reported. The study's reliance on in silico analyses, with only minimal in vitro data, limits its translational impact.
Suggestion: To strengthen the manuscript, the authors should explicitly clarify how their results offers new insights or improves upon previous studies. I recommend the inclusion of experimental validation to corroborate the computational predictions.
We sincerely thank the Reviewer for taking the time and dedication to read and evaluate our work. We are delighted with all the suggestions and we tried to address most of these. We are aware that some of the issues contemplated in this study have been previously reported, as the reviewer has noted. However, we believe that the significance of our work lies in the fact that we performed a comprehensive analysis of the potential role of RIPK1 in various relevant aspects of diffuse glioma pathology, incorporating stratification of patient biopsies according to IDH mutation status. We believe this methodology is crucial for the correct analysis of the results, since the behavior of mIDH and wtIDH gliomas is extremely different.
First of all, we found significant differences on RIPK1 expression between mIDH and wtIDH gliomas. Interestingly, patients with high RIPK1 expression showed a lower survival rate.
Consistently, our IDH mutational state stratification method allowed us to note a major correlation of RIPK1 expression and inflammatory cell death pathways, immune infiltration, proliferation and dedifferentiation related genes in wtIDH gliomas.
While it is true that some of these in silico analyses have been previously reported, many of these studies were conducted between the publication of our preprint (December 2023) and the moment in which we were able to access a waiver that would have allowed our manuscript to be considered for peer review and published if accepted. However, many of these studies failed to consider diffuse glioma databases as a whole thing, without stratifying by IDH status. In contrast, here we performed an exhaustive database study, taking account IDH status stratification.
Finally, we assessed a preliminary in vitro validation using a wtIDH GBM cell line with the aim to support our general discussion. More experiments are needed for a complete validation and we are actually working on it with a RIPK1- directed gene therapy.
Introduction comments (line 33-77)
The introduction is generally well-written and clearly outlines the study's objectives. However, it lacks a comprehensive summary of prior research investigating the association between RIPK1 expression, glioblastoma progression, and patient survival. Key studies—such as those by Dong et al. [30], Zhou et al. [31], and Melo-Lima et al. [36]—which discuss necroptosis-related gene signatures and RIPK1 activation in gliomas, are referenced only later in the Discussion and should be integrated into the Introduction to better describe the rationale for the study.
We agreed with reviewer comments about introduction. We integrated Dong et al., Zhou et al. and Melo-Lima et al. studies into the introduction and we clarified the rationale of our study (Line 79 to 122)
Additionally, important literature on the therapeutic potential of RIPK1 inhibition is absent. Several studies have evaluated RIPK1 inhibitors, alone or in combination with agents like cisplatin, and demonstrated their efficacy in promoting apoptosis and suppressing tumor proliferation in glioblastoma and other cancers (Pol et al., 2024; Zhou et al., 2020; Zheng et al., 2020; Zhang et al., 2021).
Suggestion: Incorporating these references would strengthen the scientific foundation of the manuscript and better support the proposed therapeutic relevance of RIPK1.
- Pol JG et al., RIPK1 inhibition in malignant cells potentiates immunotherapy and radiotherapy outcome. Oncoimmunology. 2024;13(1):2425465.
- Zhou J et al., Emodin induced necroptosis in the glioma cell line U251 via the TNF-α/RIP1/RIP3 pathway. Invest New Drugs. 2020;38(1):50–59.
- Zheng XL et al., RIP1 promotes proliferation through G2/M checkpoint progression and mediates cisplatin-induced apoptosis and necroptosis in human ovarian cancer cells. Acta Pharmacol Sin. 2020;41(9):1223–1233.
- Zhang Y et al., RIPK1 contributes to cisplatin-induced apoptosis of esophageal squamous cell carcinoma cells via activation of JNK pathway. Life Sci. 2021;269:119064.
We are very grateful with reviewer suggestions and all of these references were incorporated into our manuscript. (Line 79 to 122)
Methods comments (line 289-385)
The cohort size (>600 patients) is appropriate for database-driven analyses and provides sufficient statistical power to support the bioinformatic conclusions, including subgroup identification, biomarker discovery, and predictive modelling.
The only experimental validation presented involves the use of the RIPK1 inhibitor GSK2982772 to assess tumor cell growth and apoptosis under three conditions (no treatment, GSK2982772 alone, and GSK2982772 combined with 5 μM cisplatin). These assays are well executed and clearly described. However, the use of a single cell line, the absence of in vivo validation, and the lack of additional functional assays significantly limit the experimental robustness and translational relevance of the findings.
Suggestion: The authors should include one IDH-mutant and one IDH-wildtype glioma line, improving the relevance of the experimental work.
We agreed with the reviewer that more in-depth experimental validations should be performed, and that the comparison between wtIDH and mIDH cell lines would be of great relevance. However, we do not have access to human mIDH glioma cell models. We have mouse models for both types of glioma, but given the very low penetrance of our RIPK1 inhibitor across the blood-brain barrier, we opted to develop viral gene therapy vectors to address this goal. These trials are part of another ongoing project, and we are currently awaiting new funding sources to carry them out.
The primary objective of this work was to report the clinical and prognostic relevance of RIPK1 in wtIDH gliomas as quickly as possible, so that it could be considered in decision-making.
Results and Discussion comments
- Association of RIPK1 expression with de-differentiation, cell death pathways, and tumor-infiltrating immune cells (line 106-175)
The transcriptomic bioinformatic analysis confirms the association between high RIPK1 expression and pro-inflammatory and de-differentiation programs. However, these associations have already been reported in previous studies, and the current analysis—based on publicly available datasets—does not offer significant novel findings. Similarly, the correlations between RIPK1 and cell death pathways are not novel, having been described in previous studies, and are based on transcriptomic data without experimental validation. The integrated analysis of immune-related pathways is well conducted, but again, not supported by experimental evidence. Given that RIPK1’s role in immunity is already described, the novelty of all these findings is limited.
Suggestion: The inclusion of spatial transcriptomic analysis or validation in patient-derived tissues of glioblastoma would improve the association of RIPK1 expression with factors linked to activation of these pathways, providing crucial experimental validation in the tumor context.
Once again we would like to thank the Reviewer for his insightful suggestions. We incorporated another figure (Figure 7 in the new Manuscript) including Spatial Transcriptomic analyses comparing RIPK1 expression and de-differentiation, cell death pathways and tumor infiltrating cell gene signatures. Results related to Figure 7 were analyzed in line 330 to line 351and discussed between lines 526-530.
- Association of RIPK1 expression with patients survival (line 79-105, and line 183-210)
The in vitro data show that RIPK1 inhibition—particularly in combination with cisplatin—reduces cell proliferation and enhances apoptosis in glioblastoma cells. However, as previously reported, the use of a single cell line and the absence of in vivo validation significantly reduce the translational impact of these findings.
Suggestion: These limitations should be clearly acknowledged and critically discussed.
The Reviewer was correct, and we discussed the limitations of our experiments and their limited translational impact in a critical manner.(Lines 509 to 511).
Novelty
However, the study presents two original contributions that should be emphasized more by the authors:
- Stratification of RIPK1 expression by IDHwt versus IDHmut GBM
The finding that RIPK1 expression is higher in IDHwt gliomas compared to IDHmut tumors is important. This stratified analysis adds clinical relevance because IDHwt gliomas are biologically more aggressive.
- Prognostic relevance of RIPK1 expression in IDHwt versus IDHmut GBM
The observed association between high RIPK1 expression and poorer overall survival is notable in IDHwt gliomas, aligning with their more aggressive nature. This distinction between IDHwt and IDHmut tumors is the main novel aspect of the study.
Suggestion: These results should be discussed in the context of prior literature, such as Dong et al., which reported that in glioblastoma (GBM), RIPK1 expression did not correlate significantly with poorer overall and disease-free survival. The differences in RIPK1’s prognostic significance may be explained by the lack of IDH wild-type vs. mutation stratification in previous studies. Clarifying why RIPK1 prognostic impact is more pronounced in IDHwt gliomas would enhance the scientific impact of the manuscript.
Attending the Reviewer 's suggestion, we re-discussed our survival results in the context of prior literature (Line 443-447)
Figures and Tables
The figures and tables are well executed and clearly present the results of the proposed analyses.
Suggestion: in Supplementary Table 2: Add information on patient cohorts used in the survival analysis, including age, sex, IDH status, and treatment regimen.
We believe that the Reviewer is referring to Table 1. However, Table 1 already includes age, sex, IDH status information. In addition to including the number of patients who received radiation therapy, we have included information about other treatment regimens in this new version. If anything else is required, please let us know in Round 2 of Revision.
Many thanks for all of these valuable suggestions that really improved our Manuscript, and we apologise for the issues we couldn't address.
Reviewer 2 Report
Comments and Suggestions for Authors
Overall
Thank you for submitting this interesting study. I appreciate the opportunity to review your work. The following comments are provided for your consideration to help strengthen the manuscript.
Major
I noticed several instances where tracked changes with red strikethrough text remain visible in the manuscript. This raises concerns about whether all co-authors have thoroughly reviewed the manuscript and genuinely consented to its submission in its current form. I respectfully question the extent of co-author agreement regarding the submission of this manuscript.
The text within the figures is extremely small, compounded by the relatively small figure sizes overall, making them difficult to read. While I was able to review them using a large monitor with magnification, I have concerns about accessibility for readers who may be reviewing printed versions of the manuscript. This issue warrants consideration from a reader accessibility perspective.
Minor
Several formatting inconsistencies are apparent throughout the manuscript, including duplicate periods, irregular line spacing, and structural irregularities. Table 1 in section 4.1 appears to deviate from MDPI formatting guidelines.
The use of CDDP in a glioma study requires careful justification or consideration of more appropriate drug selection for this tumor type. Additionally, the control group might be better served by using the maximum volume of vehicle rather than simply "without CDDP." The study would benefit from employing multiple cell lines categorized by IDH status and WHO classification.
The inclusion of raw TUNEL data in the supplement appears unconventional and could be verified, and the manuscript lacks adequate consideration of multiple testing corrections.
The Discussion section would benefit from more comprehensive and in-depth analysis to provide greater educational value for readers. For glioma database studies, more careful attention should be paid to the evolution of WHO classifications and diagnostic criteria changes. While I understand this may be labor-intensive, it represents a crucial aspect that deserves greater consideration within the scope of this research.
I hope these comments prove helpful in strengthening your valuable contribution to the field.
Author Response
Overall
Thank you for submitting this interesting study. I appreciate the opportunity to review your work. The following comments are provided for your consideration to help strengthen the manuscript.
We are very grateful to the Reviewer for taking the time to read and critically evaluate our work. Below you will find our responses to all the comments provided.
Major
I noticed several instances where tracked changes with red strikethrough text remain visible in the manuscript. This raises concerns about whether all co-authors have thoroughly reviewed the manuscript and genuinely consented to its submission in its current form. I respectfully question the extent of co-author agreement regarding the submission of this manuscript.
We apologize to the reviewer for our mistake in uploading one of the two manuscript files with unresolved markup. However, all authors consented to the submission before and after submission. This new version only contains the additions requested by the reviewers, with track change control.
The text within the figures is extremely small, compounded by the relatively small figure sizes overall, making them difficult to read. While I was able to review them using a large monitor with magnification, I have concerns about accessibility for readers who may be reviewing printed versions of the manuscript. This issue warrants consideration from a reader accessibility perspective.
The Reviewer is right. We re-formatted figures to one page size and text in figures was adjusted too.
Minor
Several formatting inconsistencies are apparent throughout the manuscript, including duplicate periods, irregular line spacing, and structural irregularities. Table 1 in section 4.1 appears to deviate from MDPI formatting guidelines.
Formatting inconsistencies were solved following the MDPI formatting guideline.
The use of CDDP in a glioma study requires careful justification or consideration of more appropriate drug selection for this tumor type.
The rationale of using CDDP relies on that CDDP induces both DNA damage and ROS and is associated with a more inflammatory cell death than other drugs used for DG treatment such as TMZ. The use of CDDP for GBM treatment was based on several studies which combined TMZ and CDDP in clinical trials. Furthermore, Zou et al. developed nanoparticles carrying both TMZ and CDDP to avoid blood-brain-barrier with promising therapeutic potential. In fact, Zou et al. pre-clinical experiments included U251 cells. The selection of CDDP is now addressed in lines 476 to 482 and lines 486 to 488.
Additionally, the control group might be better served by using the maximum volume of vehicle rather than simply "without CDDP."
The Reviewer is right. In fact, we actually used the maximum volume of the vehicle in our controls. We now clarified that in Methods (Lines 652-653).
The study would benefit from employing multiple cell lines categorized by IDH status and WHO classification.
We agreed with the reviewer that more in-depth experimental validations should be performed, and that the comparison between wtIDH and mIDH cell lines would be of great relevance. However, we do not have access to human mIDH glioma cell models. We have mouse models for both types of glioma, but given the very low penetrance of our RIPK1 inhibitor across the blood-brain barrier, we opted to develop viral gene therapy vectors to address this goal. These trials are part of another ongoing project, and we are currently awaiting new funding sources to carry them out.
The primary objective of this work was to report the clinical and prognostic relevance of RIPK1 in wtIDH gliomas as quickly as possible, so that it could be considered in decision-making.
Limitations of our experimental validations were discussed in lines 509-511.
The inclusion of raw TUNEL data in the supplement appears unconventional and could be verified, and the manuscript lacks adequate consideration of multiple testing corrections.
Uploaded TUNEL data is not part of supplemental material of the manuscript, we submitted all the microscopy photographs corresponding to the one TUNEL experiment plotted on the figure, without any edition, as it was required during the submission of the manuscript. TUNEL assay was performed three times.
The Discussion section would benefit from more comprehensive and in-depth analysis to provide greater educational value for readers. For glioma database studies, more careful attention should be paid to the evolution of WHO classifications and diagnostic criteria changes. While I understand this may be labor-intensive, it represents a crucial aspect that deserves greater consideration within the scope of this research.
Discussion was modified attending all the suggestions of the Reviewers, we hope the new version of the manuscript meets Reviewer expectations. Discrepancies on WHO classifications and diagnostic criteria changes were solved and unified through the manuscript.
I hope these comments prove helpful in strengthening your valuable contribution to the field.
We are very grateful with the Reviewer's comments and suggestions. We think that the new manuscript including all of the valuable suggestions has been notably improved.
Round 2
Reviewer 1 Report
Comments and Suggestions for Authors
The authors have effectively addressed the suggestions and critical points raised during the review process. While some concerns remain—primarily the limited experimental validation, and secondarily the relatively few novel findings beyond the IDH mutation-based stratification—as noted in my review, the manuscript has improved considerably. In the current version, the introduction is clear and appropriately contextualized within the relevant literature, and the results/discussion now better highlight the role of IDH stratification and spatial transcriptomics in relation to RIPK1-related pathways. Notably, the integration of spatial transcriptomics in a GBM sample has added value to the study by providing additional insights into pathways activation across different neoplastic cells.
Author Response
The authors have effectively addressed the suggestions and critical points raised during the review process. While some concerns remain—primarily the limited experimental validation, and secondarily the relatively few novel findings beyond the IDH mutation-based stratification—as noted in my review, the manuscript has improved considerably. In the current version, the introduction is clear and appropriately contextualized within the relevant literature, and the results/discussion now better highlight the role of IDH stratification and spatial transcriptomics in relation to RIPK1-related pathways. Notably, the integration of spatial transcriptomics in a GBM sample has added value to the study by providing additional insights into pathways activation across different neoplastic cells.
We were delighted with the Reviewer suggestions. Our work has significantly improved by including his/her valuable comments. Although there are some issues we were not able to address, we will take it account for our future projects.
Reviewer 2 Report
Comments and Suggestions for Authors
Thank you for providing the opportunity for re-review and for your response to the previous comments. We appreciate the revisions made to address our concerns. However, upon reviewing the current version, we have identified some remaining issues that require attention. The legends and text labels in the figures appear to overlap with the graphical elements, affecting readability and clarity. There also appear to be some inconsistencies in formatting and presentation throughout the manuscript. We would appreciate it if you could review these issues and provide a corrected version. Clear and properly formatted figures are essential for effective communication of the research findings. We look forward to your revised submission.
Author Response
Thank you for providing the opportunity for re-review and for your response to the previous comments. We appreciate the revisions made to address our concerns. However, upon reviewing the current version, we have identified some remaining issues that require attention. The legends and text labels in the figures appear to overlap with the graphical elements, affecting readability and clarity. There also appear to be some inconsistencies in formatting and presentation throughout the manuscript. We would appreciate it if you could review these issues and provide a corrected version. Clear and properly formatted figures are essential for effective communication of the research findings. We look forward to your revised submission.
We downloaded the .doc file from Susy's Mdpi system and noticed that in some versions of Word, the tracked changes feature failed, causing figures to duplicate. We believe this was the problem that prevented the reviewer from viewing the file correctly. For this reason, we have uploaded the manuscript with all the changes accepted in .doc and PDF versions. We believe the figures meet the criteria previously requested by the reviewer and that this was a compatibility issue.
We are very grateful with all the Reviewer comments and suggestions. Our work has been greatly improved and we hope that this time the manuscript can be correctly visualized.